# Self-Assessment and Learning Motivation in the Second Victim Phenomenon

**DOI:** 10.3390/ijerph192316016

**Published:** 2022-11-30

**Authors:** Stefan Bushuven, Milena Trifunovic-Koenig, Michael Bentele, Stefanie Bentele, Reinhard Strametz, Victoria Klemm, Matthias Raspe

**Affiliations:** 1Institute for Infection Control and Infection Prevention, Hegau-Jugendwerk Gailingen, Health Care Association District of Constance, 78262 Gailingen, Germany; 2Institute for Medical Education, University Hospital, LMU Munich, 80336 Munich, Germany; 3Department of Anesthesiology and Critical Care, Medical Center-University of Freiburg, Faculty of Medicine, University of Freiburg, 79106 Freiburg im Breisgau, Germany; 4Training Center for Emergency Medicine (NOTIS e.V.), 78234 Engen, Germany; 5Institute for Anesthesiology, Intensive Care, Emergency Medicine and Pain Therapy, Hegau-Bodensee Hospital Singen, 78224 Singen, Germany; 6Department of Emergency Medicine, University-Hospital Augsburg, University of Augsburg, 86156 Augsburg, Germany; 7Wiesbaden Business School, Rhein Main University of Applied Sciences, 65183 Wiesbaden, Germany; 8Department of Internal Medicine, Infectious Diseases and Respiratory Medicine, Charité—Universitätsmedizin Berlin, Charitéplatz 1, 10117 Berlin, Germany

**Keywords:** second victim, clinical tribalism, overconfidence, Dunning–Kruger-effect, education, mental health

## Abstract

Introduction: The experience of a second victim phenomenon after an event plays a significant role in health care providers’ well-being. Untreated; it may lead to severe harm to victims and their families; other patients; hospitals; and society due to impairment or even loss of highly specialised employees. In order to manage the phenomenon, lifelong learning is inevitable but depends on learning motivation to attend training. This motivation may be impaired by overconfidence effects (e.g., over-placement and overestimation) that may suggest no demand for education. The aim of this study was to examine the interdependency of learning motivation and overconfidence concerning second victim effects. Methods: We assessed 176 physicians about overconfidence and learning motivation combined with a knowledge test. The nationwide online study took place in early 2022 and addressed about 3000 German physicians of internal medicine. Statistics included analytical and qualitative methods. Results: Of 176 participants, 83 completed the assessment. Analysis showed the presence of two overconfidence effects and in-group biases (clinical tribalism). None of the effects correlated directly with learning motivation, but cluster analysis revealed three different learning types: highly motivated, competent, and confident “experts”, motivated and overconfident “recruitables”, and unmotivated and overconfident “unawares”. Qualitative analysis revealed four main themes: “environmental factors”, “emotionality”, “violence and death”, and “missing qualifications” contributing to the phenomenon. Discussion: We confirmed the presence of overconfidence in second victim management competencies in about 3% of all persons addressed. Further, we could detect the same three learning motivation patterns compared to preceding studies on learning motivation in other medical competencies like life support and infection control. These findings considering overconfidence effects may be helpful for safety managers, medical teachers, curriculum developers and supervisors to create preventive educational curricula on second victim recognition and management.

## 1. Introduction

### 1.1. Background

The Second Victim Phenomenon (SVP) is a psychological reaction of health care workers (HCWs) after adverse or stressful events in medical practice and may be linked to medical error or not [1,2,3]. SVP may lead to severe psychological, physical, and social reactions and consequences for caring teams [4,5,6,7]. In times of low personal and medical resources, these factors result in further temporal or permanent loss of highly educated medical staff, lowers hospital capacities, and affect hospital economics [8], e.g., by defensive medicine [9]. Consequently, recent scientific studies, patient safety associations and political leaders emphasise the need for raising awareness, HCW training and institutional multifactorial assistance programs [10,11,12,13]. Recently our working group was able to assess the presence of SVP in German physicians and nurses [14,15], contributing to further studies on SVP in Europe [16]. However, data on clinical situations leading to SVP in Germany are rare.

In order to draw attention to SVP in HCWs, ongoing and sustainable educational programs are needed to acquire competencies in recognition and management of SVP in oneself and others [6]. Consequently, motivation for life-long learning to acquire and maintain these competencies is high. One psychological model tends to describe learning motivation based on self-determination theory [17,18]. In this theory, motivation can be divided into intrinsic motivation: “I want to learn.” and extrinsic motivation to be partitioned into identified regulation: “I have to learn, because it is my duty to do so”, extrinsic regulation: “I must learn because another person tells me to do so”, and amotivation: “I am not interested in learning”. A tool to assess these four motivation entities in learning is the SIMS (Situational Motivation Scale), originally developed for learning through physical exercise [19].

Essentially, one of the main factors in assessing time-consuming training is the recognition of learning demands by oneself or by institutional leaders. Flawed self-assessments are also known as overconfidence. The effects are described by Dunning and Kruger and their subdivision in overestimation (the belief to be better than tests reveal), over-placement (the belief of being better than others), and overprecision (the belief to be very accurate in these assumptions) by Moore [20]. A further effect is the clinical tribalism phenomenon (CTP) belonging to the in-group biases [21]: comparable to the over-placement-effect: persons estimate their own occupational group to be better than other groups. We were able to demonstrate this related effect for several infection control and communication skills in 2021, summarised in [22]. Further, our working group generated a hypothesis framework shown in Figure 1 for a better understanding of the different models and their assumed interactions.

In the last study on hand hygiene [23] and basic life support (BLS, unpublished), we identified three types of learning motivation [23]: “Experts” with justifiable confidence in their abilities, high competence and high learning motivation, “Recruitables” being overconfident and less competent but motivated and “Unawares” showed to be overconfident, less competent and not motivated to learn.

**Figure 1 ijerph-19-16016-f001:**
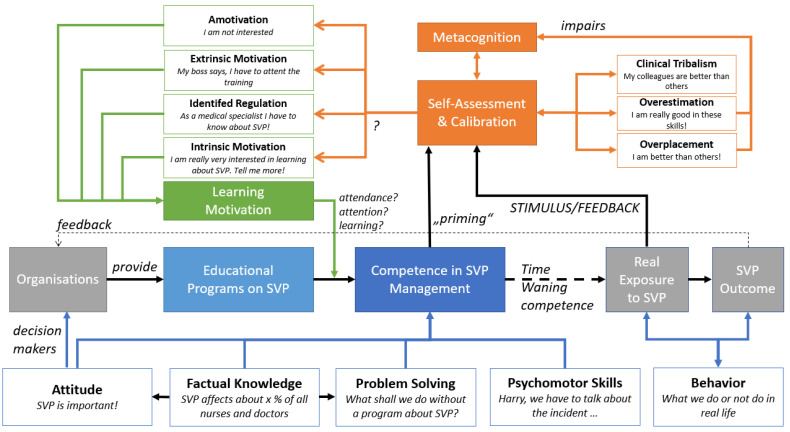
Theoretical Framework combining the different models of learning dimensions, learning motivation and overconfidence. Organisations with aware decision-makers towards SVP provide educational programs that communicate and teach attitudes, factual knowledge, problem-solving strategies and psychomotor skills [24]. Factual knowledge is important for creating attitudes and enabling problem-solving. In reality, these competencies (that may wane over time if not refreshed) are needed if an SVP situation occurs. The different behaviour of providers relying on competency may affect SVP outcome and provides modifying feedback to the health care worker. This (and maybe the first exposition to the topic, “priming”), as well as the complex issues of metacognition, may (dis)calibrate the person in his or her self-assessment on SVP competence and therefore alleviate or boost overconfidence effects [25,26] and logically the subjective need for learning. However, it remains unclear to what extent overconfidence effects contribute to learning motivation in SVP proficiency in detail (marked in the figure by “?”).

### 1.2. Rationale

The primary aim of this study is to assess the presence of overconfidence concerning competencies in second victim recognition and management and their influence on participation in training programs.

The secondary aim is to generate further hypotheses on the phenomenon based on the qualitative analysis of included open questions to determine the nature of situations leading to SVP in German hospitals as we did in preceding SeVID-Studies [14,15].

**Hypothesis** **1a** **(H1a).***Overestimation is detectable in German physicians for competencies in SVP management*.

**Hypothesis** **1b** **(H1b).***Overplacement is detectable in German physicians for for competencies in SVP management*.

**Hypothesis** **1c** **(H1c).***Clinical tribalism is detectable in German physicians for for competencies in SVP management*.

**Hypothesis** **2** **(H2).***Learning motivation to be negatively correlated to overconfidence and clinical tribal-ism*.

**Hypothesis** **3** **(H3).***Physicians could be grouped based on the assessment of their own competencies regarding SVP, factual knowledge about SVP, assessment of maximum credible risk for a victim after SVP and motivation to learn about SVP*.

This manuscript provides readers with information on overconfidence effects on SVP management, learning motivation and their intercorrelation. Additionally, it evaluated the existence of different learning types in SVP that might have to be addressed differentially by educators. Further, it provides readers with additional quantitative and qualitative data on SVP, contributing to existing evidence and context to prior studies.

## 2. Methods

### 2.1. Study Design, Setting and Participants

We designed and conducted a cross-sectional anonymous online examination in the German language on overconfidence and learning motivation. The instrument is based on prior investigations with the use of validated surveys on overconfidence and learning motivation in hand hygiene [23] and two not yet published studies on basic life support and dysphagia competencies.

After consultation with the ethical committee of the physicians’ association of Baden-Wurttemberg, Germany, we conducted the study addressing about 3000 physicians registered with the association of Internal Medicine. The survey was open to all members and was provided online from March to September 2022, comprising triple announcements via the association’s newsletter.

### 2.2. Setting

The provider of the survey platform was “umfrageonline.com”, Enuvo GmbH in Zuerich, Switzerland. IP addresses were blinded towards the investigators. Statistics and graphics were conducted using Microsoft Excel (Microsoft Corporation, Redmond, WA, USA) and SPSS 28.0 (IBM, Armonk, NY, USA).

### 2.3. Quantitative Variables and Measurement

The survey consisted of introducing information and after consent for participation of 84 items: 6 demographic, 16 SIMS, 11 self-assessments, 11 assessments on physicians, 11 assessments on nurses, 11 assessments on paramedics, 15 items testing factual knowledge and attitudes on SVP, 2 items according to ISO 31,000 risk assessments and one free text entry (see Table 1). SIMS was conducted using a 7-point Likert Scale [19] from “completely disagree” (1) to “completely agree” (7). For the assessments, we used a 5-Point Likert Scale [27], from “completely disagree” (0) to “completely agree” (4).

### 2.4. Qualitative Variables

Qualitative data was analysed in accordance with the secondary aim of the study, and data saturation was not reached. We took a phenomenological approach using a decontextualisation, coding and recontextualisation process. This was conducted by the primary investigator, who has preceding experience in qualitative analysis [28]: First, the primary investigator tagged data elements in multiple sessions by shortening the meanings of the free-text entries. He then tagged the elements the codes were formed inductively in multiple sessions as well. Third, these codes were aggregated to form the main themes. Contradictory data was not considered as it did not appear. Translation for publication was conducted secondary.

After recognising the context of emotion, we used the Ekman Model of emotion described by a consensus [29] and the “Atlas of Emotion”, taking a further deductive approach to free-text entries: We tagged the data according to the five core emotions “anger”, “fear”, “enjoyment”, “sadness” and “disgust” (see Section 3).

### 2.5. Statistical Methods

Statistics were conducted according to the preceding study [23] using Pearson’s product-moment correlation matrix, the paired sample *t*-test and quadratic regressions with bootstrapping (95% confidence interval, based on 1000 samples). In addition, we performed two-step cluster analysis for the estimation of learning motivational groups. Analysis was conducted with IBM SPSS Statistics for Windows, Version 28.0, Armonk, NY, USA).

Overestimation (H1a) was assessed descriptively by identifying the group of participants with high self-reported knowledge about SVP but failing to adequately estimate the maximum credible risk for the person experiencing SVP. A correct risk estimation was the assumption that committing suicide is of the most credible harm to a victim after SVP [30].

To display over-placement (H1b), we compared the estimations of one’s own and other providers’ (that were of the same profession) competencies in knowledge, recognition of psychological burden in oneself, speaking up and feedback reception.

Clinical Tribalism (H1c) was computed as over-placement but compared the competencies of the own professional group with those of professionals from the other two professions, namely paramedics and nurses.

Learning motivation assessed by the SIMS Score was computed by the mean of the four assigned items on intrinsic motivation, identified regulation, extrinsic motivation and amotivation each.

The correlations between over-placement and learning motivation, as well as between clinical tribalism and learning motivation, were examined using the difference in the single item of recognition of SVP in oneself and other physicians (over-placement) or the difference in the single item of recognition of SVP in oneself and other two professions (clinical tribalism) with four SIMS dimensions.

Factual knowledge about SVP was measured as an index of the cumulative score regarding adequate recognition of the potential symptoms caused by SVP. A list of 14 symptoms was presented to participants, of which only 9 can be primarily caused by SVP. The remaining five are the typical consequences of the conditions such as PTSD, Depression, Burnout or committing suicide, which can be caused by SVP.

The differences between the three learning and motivational types were estimated by two-steps cluster analysis with the four learning motivation subscales of SIMS, the mean of the differences of one’s own and others’ estimation for SVP description and recognition and the index of the cumulative score (“false” = 2 and “true” = 1) of all knowledge tests as continuous variables. Maximum credible risk to HCW, which can be caused by SVP, was added as a categorical variable in the analysis. The differentiation of the three patterns was conducted the same way as in our first study [23] to reveal if the different learning and motivational types are present in the sample. In addition, we compared the proposed cluster solution with the three cluster structures obtained in our previous studies: 1. confident experts (passing the tests), 2. overconfident recruitables (overconfident, motivated, but failing the tests) and unawares (overconfident, unmotivated and failing the tests).

## 3. Results

### 3.1. Participants and Demographic Data

Altogether 176 participants answered the questionnaire, of whom 83 (46.6%) completed the survey. Of these participants, 129 (73.3%) were female, 44 (25.08%) were male, 1 participant was non-binary (0.6%), and 2 (1.1%) gave no answer. Ages ranged from 21 to 60 years, with a mean of 33.4 years (SD 8.3 years). Of all participants, 59 were registrars, 113 were under specialised education, and six preferred not to answer. Nineteen participants were working as educators. Main working places were distributed among the participants to primary “GB-A 1”hospitals (*n* = 40, 22.7%), secondary “GB-A 2” hospitals (*n* = 32, 18.2%), tertiary “GB-A 3”hospitals (*n* = 35, 19.9%), university hospitals (*n* = 30, 17.06%), general practitioners (*n* = 28, 15.9%) and others (*n* = 11, 6.24%). Thirty-eight gave free-text comments in the comment sections.

### 3.2. Main Results

The main results and results of the cluster analysis comprising the questionnaire are shown in Table 1, Table 2 and Table 3 and Figure 2 and Figure 3.

A reassessment of the questionnaire showed a satisfying internal consistency. SIMS subscales showed an acceptable Cronbach’s Alpha (extrinsic regulation α = 0.69, intrinsic motivation α = 0.81, amotivation = 0.86, identified regulation α = 0.91) and satisfying factor allocation [CFA (χ^2^(98) = −219,031,539,276,092, *p* = 1.00, CFI = 1.00, NFI = 1.00, RMSEA = 0.00, IFI = 1.0)]. Figure 4 displays the four-factor structure of the SIMS instrument. Self-assessment of own competencies also showed good reliability (Cronbach’s Alpha = 0.84).

#### 3.2.1. H1a—Presence of Overestimation

Overestimation was assumed for persons with high confidence in their own competence but failing in correct risk stratification for SVP. We assumed the adequate risk assessment based on the single item with an estimation of the maximum credible risk of a second victim effect being potentially lethal for a health care provider [31]. Of all participants, only 29 estimated this risk correctly. Of the other persons assessing it to be of lower risk, 34.6% answered that they know about the phenomenon (fully agree or overall agree), indicating high self-reported knowledge regarding SVP. Of these persons not assessing the risk of SVP correctly, the mean result for the knowledge test was 1.90 points. In the comparison group of those assessing the risk correctly, the result for the test was 1.96 points without significance (*p* = 0.25, Cohens’ D = −0.27). In sum, both groups showed poor competencies when given the task to differentiate between the primary symptoms and the secondary symptoms, considering that 1 is the best possible score and 2 is the worst possible score. Primary symptoms can be caused by SVP, and secondary symptoms are consequences of other conditions possibly and initially caused by SVP. Thus Hypothesis 1a could be partially confirmed since we identified a considerable group of participants with highly self-assessed knowledge about SVP but failed to adequately assess the maximum risk for a person experiencing SVP and the following conditions. However, there were no differences in factual knowledge regarding the primary symptoms which can be caused by SVP between the groups of participants regarding the correctly assessed maximum risk and the incorrectly assessed maximum risk for persons affected by SVP. Hence, the formulation of the item regarding the primary symptoms of SVP: “What are Symptoms of a Second Victim phenomenon?” without emphasizing that we refer to direct primary symptoms that could have contributed to poor differentiation of the symptoms in all groups.

#### 3.2.2. H1b—Presence of Over-Placement

As well as for knowledge (see Figure 1) about SVP (*p* < 0.001, with a Cohen’s Dz of 0.54), for recognition of psychological burden (see Figure 2, *p* < 0.001 with a Cohen’s Dz of 0.88), speaking up (*p* < 0.001 with a Cohen’s Dz of 0.38) and feedback reception (*p* < 0.001, with a Cohen’s Dz of 0.96) participants rated themselves over colleagues of the same hierarchical level–confirming Hypothesis 1b.

#### 3.2.3. H1c—Presence of Clinical Tribalism

In sum, we detected significant differences between estimations of competencies for physicians in contrast to nurses and paramedics, confirming hypothesis 1b. However, the detected differences between physicians and nurses or paramedics showed a Cohen’s D of 0.59–0.91 and 0.47–0.51, respectively.

#### 3.2.4. H2—Correlation of Learning Motivation and Overconfidence Effects

There was no linear or quadratic correlation between learning motivation and overconfidence effects (*p* > 0.05), rejecting hypothesis 2.

#### 3.2.5. H3—Detection of Three Learning Motivation Types

The Two-Step Cluster analysis for 80 completers of the survey revealed the existence of three groups:Group “A” comprising 35 persons, showed a “critical risk” estimation, low amotivation (M = 2.12), lowest Self-Assessment (M = 2.04), high intrinsic motivation (M = 5.27), high identified regulation (M = 6.28), medium extrinsic motivation (M = 2.69) and 1.93 indexes in the tests.Group “B” with 29 persons, showed a “lethal risk” estimation, low amotivation (M = 1.75), highest Self-Assessment (M = 2.67), high intrinsic motivation (M = 5.45), high identified regulation M = 6.11 medium extrinsic motivation (M = 2.81) and 1.96 indexes in the tests.Group “C” with 16 persons, showed a “moderate risk” estimation, higher amotivation (M = 2.94), high Self-Assessment (M = 2.31), lower intrinsic motivation (M = 4.61), moderate identified regulation M = 5.50, medium extrinsic motivation (M = 2.73) and 1.83 indexes in the tests.

Compared to our preceding studies in hand hygiene and basic life support, the three clusters are again assignable to the groups of “Recruitables” (group A), “Experts” (group B) and “Unawares” (group C), confirming Hypothesis 3.

### 3.3. Qualitative Findings

Altogether 36 free-text entries on Q 57 and 26-free-text entries on Q58 could be analysed in what situations participants experienced SVP and what happened after that (Q57: In what situations did you experience Second Victims phenomena? What were the results of that?”) and whether participants may have further suggestions for the survey or want to report on further experiences with SVP (Q58: “Do you have any suggestions, special experiences or comments about the Second Victim Phenomenon?”. Qualitative entries included 1933 words for both questions.

Some did confirm their experience in SVP or did not further specify their experiences: “*I do not want to mention the situation. But I was incapable of working for days*” and “*At work*” were some of these answers.

After decontextualization and recontextualization, major preliminary themes for the first questions were the following

1.Environmental and Institutional Factors

Participants reported on the occurrence of SVP to their experience in several locations in and out-of-hospital, especially in Emergency departments, Emergency medicine outside hospitals and, in particular, endoscopy. The first code comprises the **environmental and institutional setting** itself with environmental, structural and team factors resulting in the following phrases translated from German into English: “*Dying child in out-of-hospital emergency service*”, “*Child resuscitations in the ED*”, “*Obese patient outside the hospital. Initially there was no suitable ambulance car for him so that he deteriorated and died later in the hospital*”, “*There was no counselling during ‘praep-course’ [anatomy course]*”, “*it happened in the endoscopy room in an emergency situation without backup during on-call duty*”, “*There was a person under resuscitation each team tried to shift to the other team*”, “*There was a dying patient at home who refused further therapy*”, “*I missed relevant tests after stressful hours of work on-call*”. A second code comprised several times **leaving the medical institutions with aggravation of the situation for remaining staff (“dropping out”)**:

“*I left emergency medicine after that*”, “*It hits those colleagues with moral beliefs harder, leading to them fleeing the hospital*. Ad hoc *quitting leads to further staff shortage making colleagues consequently thinking about quitting too. And those colleagues staying and who are not morally affected are promoted*.” “*I consulted my doctor and my attorney. After that I decided to sign off sick. After that I was fired. Now I am looking for a job, but I am cancelling every job interview due to my anxiety*”, “*I had it in the ED. After several times I changed my workplace*”, “*I am anxious to work further in endoscopy, so I moved on into a practice outside the hospital*”, “*I witnessed a dislocation of an ECMO-cannula. After that I cannot smell coagulated blood anymore […] and consequently I moved to a field without emergency medicine*”, “*I am experiencing a distance from my profession and I would not recommend this job to my children*”, “*After such events I feel no joy in my work anymore*”.

*The third code was***Leadership Culture** aggravating or inflicting the SVP: “*There was an error, and the supervisor let my colleague down*”, “*Interpersonal factors in teams, especially with newbies*”, “*Permanently tolerated under-staffed situation due to economical calculations*”, “*No safety management in the hospital*”, “*My supervisor was violent*”, “*I got in the ED without onboarding*”, “*I fear my supervisors*”, “*There are conflicts among colleagues if they are understaffed*”, “*I was blamed by the prehospital emergency physician*”.

2.Adverse situations generate emotions and feelings of guilt in the Second Victims

In this second theme and after recognizing the high emotional load, we used a deductive approach according to the consensus on emotion described by Paul Ekman [29]: In this scientific consensus about human emotions, these are divided into the five basic emotions anger, fear, disgust, enjoyment and sadness with different intensity. We chose this deductive approach for better comparability with other research in medicine and the established use of coping mechanisms and communication techniques (e.g., mnemonics) addressing emotions known to physicians and that may not only help to support patients but colleagues too [32,33].

The main codes were “**Anger**”, “**Fear**”, and “**Loss of Emotions**”. The other emotions (enjoyment, disgust and sadness) were not or rarely detected: “*There was an error, and the supervisor let my colleague down*” (see above), “*There are colleagues suffering from panic and anxiety until quitting the job*”, “*Reduction of empathy*”, “*Cynicism*”, “*You are angry whenever that happens*”, “*I am full of fear*”, “*I am fearing on-call duties*.”, “*I am afraid of committing mistakes and getting rebuked for it*”, “*I am fearing my supervisor*”, “*I am afraid that something bad happens to my patient*.”, “*I am fearing a patient’s death*”, “*There is much anger among colleagues*”, “*I fear that this could happen again*”, “*I do not want to be alone on a ward anymore*”, “*I am afraid to get involved again in a paediatric emergency case*”, “*I was feeling incompetent*”, “*I witnessed a resuscitation of a child as young as mine*”, “*I am afraid to treat somebody after that event*”, “*I fear my job*”, “*I feel guilt–massively*”, “*I am afraid of similar situations happening, thus I overtreat patients*”.

3.Contributing patient factors to SVP are violence and life-threatening situations

The two codes of this theme (**Death** and **Violence**) were mentioned by several responders, and most of the phrases could be coded multiple times (see Themes 1 and 2). However, death or life-threatening situations played a significant role in the results, comprising a third subcode named “COVID”: “*Unexpected death of young patients*”, “*Intrauterine death with dreams about that child for weeks*”, “*The whole COVID time with the death of young people*”, “*Death of a child during out-of-hospital in emergency service*”, “*During the pandemic. Especially in severe cases of patients with futile condition*”, “*pediatric CPR*”, “*In the post-mortem examinations*”, “*In futile conditions*”, “*Unexpected CPR after CT-scan*” were mentioned for the code death. For “violence”, responders stated phrases like “*Violence by patients, especially distance-less drug-dependent men*”, “*Violence by policemen with fear for repetition of the misuse of power*”, “*Violence towards patients*”, “*Conflicts with family members*”, “*A patient insulted me and also infected me with varicella. Now I fear to get shingles myself*”, “*Patients were very unsatisfied and angry about my medical decisions*”.

4.Missing qualifications and experience lead to the second victim phenomena

For the last theme and in addition to similar phrases in other themes, respondents mentioned typical phrases showing overexertion in critical environments, e.g., “*there are low competencies in the ED*”, “*lack of experience*”, and in an assumed palliative case, “*there was no possibility to ventilate a dying patient or to mitigate his symptoms*”.

In the second question, we asked for possible interventions to compensate for or prevent SVP. Most participants mentioned, apart from compensation for staff shortage, structural changes in the hospital with intervention programs and deeper education on SVP to be necessary: “*regular assessments of physical and psychological conditions*”, “*assessment of competencies*”, “*Caring for students*”, “*More awareness and education with an open mind for emotion and feelings. Hospitals should stop to be a shark tank. And we should learn that we are not perfect.*”, “*We need supervisions […] and conflict training for supervisors. We are in a balancing act between self-care and care for psychically ill and violent patients*”, “*open-minded management of errors*”, “*We need clear onboarding concepts and tutors not merely on the paper.*”, “*Education of supervisors before they advance to leading positions and supervisions by independent persons*”, “*Structured educational programs and compulsory psychological supervisions*”, “*Errors must not be a taboo*”, “*peer-support*”, “*We need partners we can speak to and regular supervision*”, “*The issues have to be part of the education*”, and “*balint groups*” were some to the mentions.

## 4. Discussion

### Key Results

To our best knowledge, this is the first study reporting on the interdependency of overconfidence effects (overestimation and over-placement) and learning motivation concerning the second victim phenomena.

We were able to show that overestimation and over-placement for parts of the proficiency of second victim management are detectable in a closed population of physicians of internal medicine (H1a, H1b). To our knowledge, there is no further evidence for the detection of overconfidence in SVP management as it is described for other psychological or psychiatric entities in which limited self-insight is a known phenomenon [34,35].

Clinical tribalism or a linear or quadratic correlation between overconfidence effects and learning motivation could not be detected (H1c, H2). This is contrary to our prior findings for hand hygiene [36] and basic life support [unpublished] using the same instruments and statistical analyses. However, the differences in rating oneself and others in SVP are much lower than for hand hygiene [23] as a very common and likely easier task than dealing with a complex psychological entity most physicians are not trained for. This can also be seen in the cluster analysis: Although we were able to confirm the three groups as in hand hygiene and in BLS, the differences between the groups were smaller compared to the preceding studies. This might be due to either the novelty of the SVP concept [14,15] or the complexity compared to BLS and hand hygiene. Therefore, it might be possible that with rising interest in SVP and implementation of national learning objective catalogues and evolving awareness in all medical professional groups, we might face a “polarisation” or “differentiation” of the three clusters comparable to other, “older” or “more common” proficiencies. However, it is speculative whether “young” medical proficiencies compared to “mature” proficiencies differ in learning motivation or whether general safety culture might play a role in learning motivation and motivation to conduct learning skills [37].

Next, we could show that in the absence of “horizontal” clinical tribalism (among the professions [21]), it might be possible that there is a kind of “vertical” clinical tribalism towards students and supervisors. While students may be found easy to be cared for, our data showed that responders seem to not speak up on supervisors in case of witnessing SVP symptoms in them–likely based on hierarchical issues and perhaps anxiety about being blamed for an insulting intervention and fear for stigmatisation or loss of authority [22,38,39]. Especially for leaders, it might be relevant that if they face SVP or even develop psychic pathologies, this may affect the safety of patients, staff and the institution’s reputation and economy [2]. Consequently, the education of leaders/role models is important to de-stigmatise SVP on the one hand and to lower barriers to speaking up on the other.

A further finding of our analysis is that emotions like anger and anxiety may play a significant role in SVP, indicating the need for the implementation of briefing, debriefing [40], counselling or peer support [41] with the potential use of established communication strategies like the NURSE or SPIKES mnemonics for patients [32,33] that may have to be evaluated in future studies for the use on colleagues. Further, research should concentrate on the emotional impact as emotions and feelings play a significant role and may be a barrier in communication and speaking up–- that is difficult even if the person giving feedback is calm and sentient. Consequently, education on SVP recognition and management must comprise teaching of communication de-escalation skills not only in advanced life support courses [40,42] like ALS, ACLS, PALS or ACiLS [43] but moreover as a life-long learning issue for all medical professionals and parts of the chain of rescue from out-of-hospital to the ED, the cath lab, endoscopy or surgical ward, to high- and medium dependency units (ICU, IMC) and rehabilitation hospitals. With regard to our findings for learning motivation, suitable and attractive courses for all kinds of learners have to be considered, addressing or even inspiring the unawares.

A further issue in the qualitative analysis was violence by patients or even colleagues. This is consistent with our findings in German nurses in the SEVID II Study [14]. Interestingly, in SEVID I, which was conducted in the same population as our study but earlier, violence was not mentioned often, indicating that longitudinal research to rule out a selection bias and to detect environmental and social circumstances that may aggravate violence in hospital settings along the pandemic.

As this study was conducted on internal medicine physicians, we do not have deeper information on other occupational medical groups like surgeons, paediatricians, obstetricians or anesthesiologists or other professions (nurses, paramedics, midwives, etc.) in Germany. However, in other countries, the effect on these groups has been described earlier [2,44,45,46,47,48].

## 5. Limitations

This study faces several limitations to be considered for the use of the results and interpretation:

Selection bias is the most dominant bias in all surveys. Despite using a closed population of about 3000 individuals in Germany, only 176 responded, and 83 completed the survey. This comprises only about 2.7% of the addressed population and limits representativeness despite good validity concerning the demographic data. We concluded that the addressed physicians face high workloads in the pandemic (e.g., represented by the DGINA ample, a tool assessing workload in emergency departments in Germany [49]) that could lower newsletter readings and, therefore, participation. Further, the situation in this group addressed in other countries is not known for certain, and pre-pandemic data might differ from pandemic data. Moreover, it might be possible that persons suffering from SVP or even posttraumatic stress disorder (PTSD) (as some answers in the qualitative section strongly suggest) may be more motivated to participate and to report on their experiences. Therefore, further researchers might concentrate on smaller closed populations and measurements of PTSD symptoms.

Second, the response burden of this rather long survey might play a role in the preselection of participants. The long survey and motivational factors mentioned above might lead to an overrepresentation of those persons suffering from PTSD and SVP with motivation to display their experience and feelings. Under consideration of the growing staff shortage in German medicine and alarming demographics, every physician (after time- and resource-intensive education) contributes to the medical system and must not be overlooked.

Third, recall bias might play a certain role in the qualitative data retrieved, as experiences with highly emotional content might change with time. Further research may concentrate on the question of whether the perception of adverse events and memory concerning them is robust or may change over time.

A fourth main limitation is that our questionnaire was not designed to test the whole competence of second victim management. We were only able to test for the learning dimensions of knowledge and attitude as surrogate parameters but not for proficiency in psychomotor skills (like speaking up), problem-solving (depending on the structure of the workplace) or even behaviour in daily activities. These learning dimensions [24] may be very difficult to validly assess as they would need a full-time assessment that would be biased again by observation, as can be seen in other competencies [50]. Nevertheless, it might be possible to re-test our hypotheses for some of the learning dimensions in simulation settings.

The last limitation is the analysis of the qualitative findings. Although they show certain points of view and may contribute to further hypothesis generation, we did not reach data saturation (that was not the intention of this study). Thus, we were only able to get the first glimpse of qualitative SVP aspects in Germany and declared these qualitative findings to be preliminary. They may contribute to ongoing [51] and future qualitative studies on SVP. Additionally, it might be important that qualitative studies in this field should not rely on text interpretation alone but possibly on paraverbal and expressive data too, e.g., by videotaped interviews and especially coding done by more than one researcher.

## 6. Translation of the Results

The results show either overestimation and over-placement effects in different groups of physicians but not clinical tribalism effects. Further, we detected emotional responses that are in line with preceding studies on the psychosocial effects of traumatic events in health care, emphasising the need for educational programs on SVP and the establishment of self-care and self-help in case of SVP and for communication strategies for peers and supervisors in case of strong emotional reaction before professional help can start: probably a “First Aid concept on Second Victims and Wounded Healers” directly on the scene. Further, we were able to show that the distinction in the three learning groups, as shown in hand hygiene, was not that prominent in SVP, possibly due to the “novelty” of the term and concepts and low frequency in training on this issue. Thus, learning motivation may be high in “naïve” providers when hospitals and institutions are starting programs, but hypothetically may be followed by a “split-up” and more polarization into the three groups with the creation of an amotivational group not easily motivated to attend training to acquire or maintain proficiencies in SVP recognition and support. Future studies should concentrate on this issue after a certain time the implementation of SVP programs or even better by longitudinal observation of self-assessment and learning motivation.

## 7. Conclusions

In this study, we were able to detect overestimation and over-placement effects for SVP knowledge and recognition competencies. Despite a low response rate, these findings might help supervisors and medical educators in the development of curricula and training units about SVP and coping mechanisms. Further, our results may be useable for peer supporters, supervisors, tutors and mentors to identify traumatising situations and to give support to second victims. SVP researchers may use some of the data as stated above for ongoing [51] and future investigations. For researchers in medical didactics, the evaluation of the impact of overconfidence effects on learning motivation in different medical proficiencies may be useful as this and related effects might affect learning motivation in some but not all persons eligible for training.

Further, we were able to show that the three learning clusters of experts, recruitables and unawares exist but are not that differentiated from in hand hygiene and basic life support. Further efforts should concentrate on medical education research on SVP and the longitudinal development of the effects.

## Figures and Tables

**Figure 2 ijerph-19-16016-f002:**
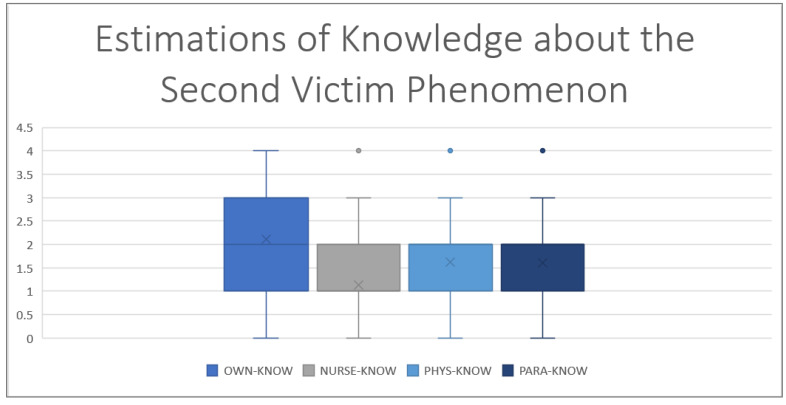
Estimations of one’s own and others’ knowledge (KNOW) about SVP for own, nurses (NURSE), physicians (PHYS) and paramedics (PARA). The *Y*-axis displays points according to the Likert scale with a minimum of 0 and a maximum of 4, showing the over-placement of participants above all other professional groups.

**Figure 3 ijerph-19-16016-f003:**
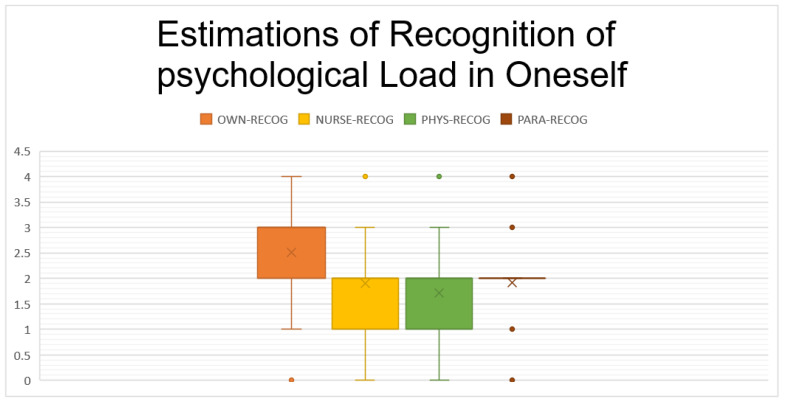
Estimations of one’s own and others’ ability to detect SVP in oneself, for own nurses (NURSE), Physicians (PHYS) and Paramedics (PARA) competencies. *Y*-axis displays points according to the Likert scale with a minimum of 0 and a maximum of 4. Again, respondents rated their own competencies higher than others’ competencies.

**Figure 4 ijerph-19-16016-f004:**
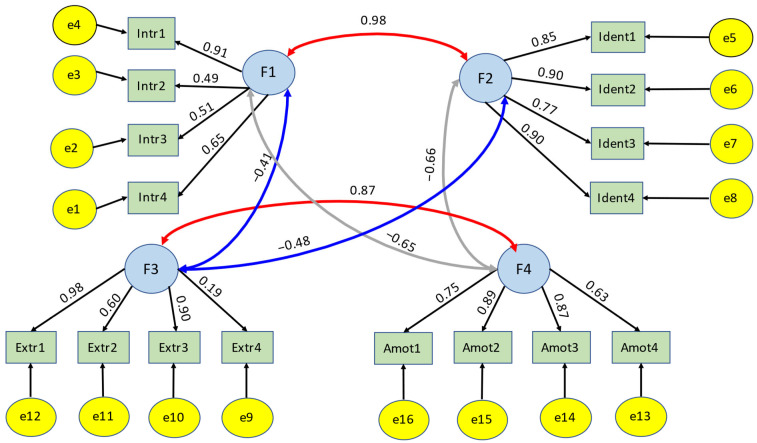
Four-factor structure of the SIMS instrument for validation of the instrument, standard regressions estimates are on the paths; Intr1, ntr2, Intr3, Intr4 variables measuring internal motivation, Extr1, Extr2, Extr3, Extr4 manifest variables measuring external regulation, ident1, ident2, ident3, ident4 manifest variables measuring identified regulation and Amot1, Amot2, Amot3, Amot4 manifest variables measuring amotivation.

**Table 1 ijerph-19-16016-t001:** Questionnaire and the results for all participants and the subgroups (M = mean, SD = standard deviation). Questions 8 to 51 used a 5-Point Likert Scale (“completely agree” = 4 to “completely disagree” = 0). Item 7 (the SIMS) used a 7-point Likert scale (“completely agree” = 7 to “completely disagree” =1). Other item scales are shown in the table. ALL are all participants who responded to the item, not necessarily completing the questionnaire, i.e., the participants who were not included in the cluster analysis due to non-response to all relevant items for the cluster analysis are included in ALL statistics. % of ALL is the proportion of all participants who responded to the item, and % of EXPERTS, RECRUITABLES and UNAWARES is the proportion of participants in the cluster.

	All 81 < *n* < 177	Experts *n* = 29	Recruitables *n* = 35	Unawares *n* = 16
Gender	Male: 44 (25%) Female: 129 (73.3%) Non-binary: 1 (0.6%) Not specified: 2 (1.1%)	Male: 4 (13.8%) Female: 25 (86.2%)	Male: 9 (25.7%) Female: 25 (71.4%) Not specified: 1 (2.9%)	Male: 5 (31.3%) Female: 11 (68.8%)
Mean Age	M = 33.40 SD = 10.00	M = 36.62 SD = 12.26	M = 34.54 SD = 10.96	M = 33.81 SD = 3.19
Educator status	19 (10.8%)	4 (13.8%)	2 (5.7%)	2 (12.5%)
Intrinsic Motivation	M = 5.09 SD = 1.06	M = 5.45 SD = 1.07	M = 5.26 SD = 0.83	M = 4.64 SD = 1.08
Identified regulation	M = 5.87 SD = 1.05	M = 6.28 SD = 0.85	M = 6.11 SD = 0.69	M = 5.50 SD = 0.97
Extrinsic Motivation	M = 2.92 SD = 1.18	M = 2.81 SD = 1.10	M = 2.59 SD = 0.85	M = 2.95 SD = 0.94
Amotivation	M = 2.48 SD = 1.28	M = 1.75 SD = 0.81	M = 2.11 SD = 0.73	M = 2.97 SD = 1.45
Q.8: I am able to describe the concept “second victim”	M = 2.12 SD = 1.27	M = 2.83 SD = 1.00	M = 1.91 SD = 1.07	M = 1.81 SD = 1.23
Q.9: I am competent to detect signs of psychological stress in myself	M = 2.51 SD = 0.86	M = 2.59 SD = 0.83	M = 2.34 SD = 1.00	M = 2.44 SD = 1.03
Q.10: I am competent to detect signs of psychological stress in other physicians	M = 2.26 SD = 0.816	M = 2.17 SD = 0.76	M = 2.20 SD = 0.87	M = 2.31 SD = 1.01
Q.11: I am competent to detect signs of psychological stress in nurses	M = 2.15 SD = 0.81	M = 2.00 SD = 0.89	M = 2.06 SD = 0.73	M = 2.38 SD = 0.89
Q.12: I am competent to detect signs of psychological stress in paramedics	M = 1.73 SD = 0.98	M = 1.76 SD = 1.02	M = 1.60 SD = 0.91	M = 1.69 SD = 1.01
Q.13: I am competent to detect signs of psychological stress in students of my profession	M = 2.11 SD = 0.87	M = 2.07 SD = 0.96	M = 2.03 SD = 0.82	M = 2.00 SD = 0.89
Q.14: I am competent to detect signs of psychological stress in supervisors of my profession	M = 1.97 SD = 0.91	M = 1.90 SD = 0.90	M = 1.86 SD = 0.84	M = 2.06 SD = 1.06
Q.15: I am addressing students whenever I detect signs of psychological stress in them	M = 2.02 SD = 1.17	M = 2.28 SD = 1.19	M = 1.86 SD = 1.19	M = 2.00 SD = 1.27
Q.16: I am addressing colleagues of my hierarchical level whenever I detect signs of psychological stress in them	M = 2.27 SD = 1.10	M = 2.31 SD = 1.26	M = 2.09 SD = 1.07	M = 2.50 SD = 1.21
Q.17: I am addressing supervisors whenever I detect signs of psychological stress in them	M = 1.08 SD = 0.99	M = 0.97 SD = 0.98	M = 1.00 SD = 0.94	M = 1.13 SD = 0.81
Q.18: I accept feedback appropriatelyif I am addressed by another person in case of psychological stress.	M = 2.52 SD = 0.90	M = 2.45 SD = 0.95	M = 2.66 SD = 0.91	M = 2.56 SD = 0.89
Q.19: Nurses know the concept of “Second Victim”	M = 1.14 SD = 0.87	M = 1.18 SD = 0.81	M = 1.11 SD = 0.80	M = 0.88 SD = 0.62
Q.20: Nurses are competent to detect signs of psychological stress in themselves.	M = 1.90 SD = 0.86	M = 1.34 SD = 0.90	M = 1.77 SD = 0.84	M = 1.88 SD = 0.81
Q.21: Nurses are competent to detect signs of psychological stress in physicians.	M = 1.75 SD = 0.93	M = 1.90 SD = 0.94	M = 1.74 SD = 0.89	M = 1.94 SD = 1.12
Q.22: Nurses are competent to detect signs of psychological stress in nurses.	M = 2.32 SD = 0.85	M = 1.72 SD = 0.92	M = 2.20 SD = 0.68	M = 2.25 SD = 1.00
Q.23: Nurses are competent to detect signs of psychological stress in paramedics.	M = 1.52 SD = 0.97	M = 2.45 SD = 0.95	M = 1.46 SD = 0.92	M = 1.44 SD = 1.15
Q.24: Nurses are competent to detect signs of psychological stress in students of their own professions.	M = 2.00 SD = 0.81	M = 1.55 SD = 0.99	M = 2.00 SD = 0.69	M = 2.06 SD = 1.00
Q.25: Nurses are competent to detect signs of psychological stress in supervisors of their own profession.	M = 1.68 SD = 0.96	M = 2.00 SD = 0.85	M = 1.60 SD = 0.88	M = 1.88 SD = 1.20
Q.26: Nurses address Students if they detect signs of psychological stress in them	M = 1.92 SD = 0.83	M = 1.86 SD = 0.92	M = 1.94 SD = 0.68	M = 1.88 SD = 0.72
Q.27: Nurses address colleagues of their own hierarchical level if they detect signs of psychological stress in them	M = 2.05 SD = 0.83	M = 1.86 SD = 0.92	M = 2.06 SD = 0.77	M = 2.00 SD = 0.97
Q.28: Nurses address supervisors if they detect signs of psychological stress in them	M = 1.23 SD = 0.76	M = 1.97 SD = 0.78	M = 1.17 SD = 0.71	M = 1.19 SD = 0.66
Q.29: Nurses accept feedback appropriately if they are addressed by another person in case of psychological stress.	M = 1.92 SD = 0.79	M = 1.31 SD = 0.71	M = 1.80 SD = 0.76	M = 1.81 SD = 0.66
Q.30: Physicians know the concept of “Second Victim”	M = 1.63 SD = 0.97	M = 2.07 SD = 0.84	M = 1.51 SD = 0.92	M = 1.31 SD = 0.87
Q.31: Physicians are competent to detect signs of psychological stress in themselves.	M = 1.71 SD = 0.77	M = 1.93 SD = 0.84	M = 1.71 SD = 0.75	M = 1.56 SD = 0.81
Q.32: Physicians are competent to detect signs of psychological stress in physicians.	M = 1.91 SD = 0.78	M = 1.72 SD = 0.80	M = 1.83 SD = 0.79	M = 1.88 SD = 0.89
Q.33: Physicians are competent to detect signs of psychological stress in nurses.	M = 1.78 SD = 0.77	M = 1.97 SD = 0.63	M = 1.71 SD = 0.79	M = 1.63 SD = 0.81
Q.34: Physicians are competent to detect signs of psychological stress in paramedics.	M = 1.60 SD = 0.80	M = 1.86 SD = 0.64	M = 1.71 SD = 0.71	M = 1.25 SD = 0.78
Q.35: Physicians are competent to detect signs of psychological stress in students of their own professions.	M = 1.82 SD = 0.76	M = 1.66 SD = 0.81	M = 1.77 SD = 0.65	M = 1.56 SD = 0.81
Q.36: Physicians are competent to detect signs of psychological stress in supervisors of their own profession.	M = 1.68 SD = 0.80	M = 1.76 SD = 0.74	M = 1.63 SD = 0.84	M = 1.44 SD = 0.73
Q.37: Physicians address Students if they detect signs of psychological stress in them	M = 1.75 SD = 0.95	M = 1.83 SD = 0.71	M = 1.74 SD = 0.92	M = 1.56 SD = 1.15
Q.38: Physicians address colleagues of their own hierarchical level if they detect signs of psychological stress in them	M = 1.81 SD = 0.91	M = 1.90 SD = 0.86	M = 1.77 SD = 0.84	M = 1.88 SD = 1.20
Q.39: Physicians address supervisors if they detect signs of psychological stress in them	M = 1.03 SD = 0.76	M = 1.10 SD = 0.72	M = 1.03 SD = 0.71	M = 0.75 SD = 0.58
Q.40: Physicians accept feedback appropriately if they are addressed by another person in case of psychological stress.	M = 1.71 SD = 0.73	M = 1.72 SD = 0.65	M = 1.77 SD = 0.69	M = 1.50 SD = 0.82
Q.41: Paramedics know the concept of “Second Victim”	M = 1.60 SD = 0.90	M = 1.76 SD = 0.79	M = 1.57 SD = 0.92	M = 1.19 SD = 0.83
Q.42: Paramedics are competent to detect signs of psychological stress in themselves.	M = 1.92 SD = 0.81	M = 1.83 SD = 0.66	M = 2.00 SD = 0.84	M = 1.75 SD = 0.93
Q.43: Paramedics are competent to detect signs of psychological stress in physicians.	M = 1.60 SD = 0.84	M = 1.69 SD = 0.71	M = 1.51 SD = 0.85	M = 1.56 SD = 0.89
Q.44: Paramedics are competent to detect signs of psychological stress in nurses.	M = 1.57 SD = 0.81	M = 1.69 SD = 0.66	M = 1.51 SD = 0.82	M = 1.38 SD = 0.89
Q.45: Paramedics are competent to detect signs of psychological stress in paramedics.	M = 2.08 SD = 0.87	M = 2.07 SD = 0.80	M = 2.09 SD = 0.89	M = 1.94 SD = 1.06
Q.46: Paramedics are competent to detect signs of psychological stress in students of their own professions.	M = 1.83 SD = 0.86	M = 1.83 SD = 0.76	M = 1.80 SD = 0.87	M = 1.88 SD = 0.96
Q.47: Paramedics are competent to detect signs of psychological stress in supervisors of their own profession.	M = 1.72 SD = 0.82	M = 1.86 SD = 0.69	M = 1.63 SD = 0.77	M = 1.63 SD = 0.96
Q.48: Paramedics address Students if they detect signs of psychological stress in them	M = 1.90 SD = 0.87	M = 1.90 SD = 0.67	M = 1.83 SD = 0.89	M = 2.00 SD = 1.10
Q.49: Paramedics address colleagues of their own hierarchical level if they detect signs of psychological stress in them	M = 1.94 SD = 0.86	M = 1.97 SD = 0.73	M = 1.94 SD = 0.91	M = 1.81 SD = 1.05
Q.50: Paramedics address supervisors if they detect signs of psychological stress in them	M = 1.48 SD = 0.86	M = 1.55 SD = 0.69	M = 1.37 SD = 0.91	M = 1.44 SD = 0.89
Q.51: Paramedics accept feedback appropriately if they are addressed by another person in case of psychological stress.	M = 1.77 SD = 0.75	M = 1.73 SD = 0.65	M = 1.83 SD = 0.79	M = 1.63 SD = 0.81
Q52: What are symptoms of a Second Victim phenomenon (True/False)				
Emotional Reactions (TRUE)	79 (99%)	28 (96.6%)	35 (100%)	16 (100%)
Absenteeism (TRUE)	72 (90%)	28 (96.6%)	30 (85.7%)	14 (87.5%)
Change of World View (FALSE)	8 (10%)	3 (10.3%)	3 (8.6%)	2 (12.5%)
Sleeplessness (TRUE) Schlaflosigkeit	79 (100%)	28 (96.6%)	35 (100%)	16 (100%)
Reduction of contacts to friends (TRUE)	75 (94%)	27 (93.1%)	33 (94.3%)	15 (93.2%)
Psychological Stress (TRUE)	80 (100%)	29 (100%)	35 (100%)	16 (100%)
Physical Stress (TRUE)	80 (100%)	29 (100%)	35 (100%)	16 (100%)
Suicidal thoughts (FALSE)	9 (11.2%)	2 (6.9%)	3 (8.6%)	4 (25%)
Cynicism (FALSE)	3 (3.8%)	2 (6.9%)	1 (2.9%)	0 (0%)
Avoidance of risky activities (TRUE)	75 (76.3%)	27 (93.1%)	32 (91.4%)	16 (100%)
Feelings of Guilt (TRUE)	80 (100%)	29 (100%)	35 (100%)	16 (100%)
Change of the religion (FALSE)	28 (37%)	10 (34.5%)	15 (42.9%)	3 (18.8%)
Alcoholism (FALSE)	4 (2.6%)	2 (6.9%)	2 (5.7%)	0 (0%)
Permanent Fatigue (FALSE)	3 (3.8%)	0 (0%)	2 (5.7%)	1 (6.3%)
Q.53: The maximum credible harm to a health care provider suffering from the second effect is…				
…insignificant	0 (0%)	0 (0%)	3 (8.6%)	0 (0%)
…minor, with short-term inability to work but without permanent harm	1 (1.25%)	1 (100%)	10 (28.6%)	0 (0%)
…major, with recurrent episodes of sick leaves	15 (18.8%)	0 (0%)	13 (37.1%)	15
…critical, with permanent physical or psychological Damage and permanent incapacity for work	35 (44.8%)	0 (0%)	7 (20%)	1 (2.9%)
…catastrophic, with severe psychological burden up to committing suicide	29 (36.2%)	0 (0%)	2 (5.7%)	0 (0%)
…major, with recurrent episodes of sick leaves	15 (18.8%)	0 (0%)	13 (37.1%)	15
…critical, with permanent physical or psychological Damage and permanent incapacity for work	35 (44.8%)	0 (0%)	7 (20%)	1 (2.9%)
…catastrophic, with severe psychological burden up to committing suicide	29 (36.2%)	0 (0%)	2 (5.7%)	0 (0%)
Q.54: How often is it in your environment for a health care provider to experience this estimated harm?				
…less than once in 3 years	8 (8.7%)	3 (10.3%)	3 (8.6%)	2 (12.5%)
…more frequent than once in 3 years	22 (23.9%)	4 (13.8%)	10 (28.6%)	6 (30%)
…more frequent than once in 1 year	32 (34.7%)	9 (31%)	13 (37.1%)	6 (37.5%)
…more frequent than once in 3 months	17 (18.5%)	10 (34.5%)	7 (20%)	0 (0%)
…more frequent than once in 1 month	5 (18.4%)	3 (10.3%)	2 (5.7%)	8 (50%)
Q.55: The maximum credible harm to a patient cared for by a health care provider suffering from the second victim effect is…?				
…insignificant	0(0%)	0 (0%)	0 (0%)	0 (0%)
…minor, with short-term inability to work but without permanent harm	8 (10%)	0 (0%)	13 (10.3%)	0 (0%)
…major, with harm with need for longer care	36 (45%)	9 (31.03%)	17 (48.6%)	5 (31.3%)
…critical, with permanent physical or psychological Damage	19 (23.8%)	7 (24.14%)	11 (31.43%)	10 (62.5%)
…lethal	25 (31.3%)	12 (41.4%)	13 (81.3%)	1 (6.3%)

**Table 2 ijerph-19-16016-t002:** Results of paired samples t-tests with bootstrapping (bias-corrected and accelerated (BCa) based on 1000 samples) of differences between the assessment of physicians’ and nurses’ competencies estimated by physicians.

Item	Physician	Nurse	*p*, Dz
… know the concept of “Second Victim.”	M = 1.63 SD = 0.97	M = 1.14 SD = 0.87	*p* < 0.001; Dz = 0.62
… are competent to detect signs of psychological stress in themselves.	M = 1.71 SD = 0.77	M = 1.90 SD = 0.86	*p* = 0.04; Dz = 0.62
… are competent to detect signs of psychological stress in physicians.	M = 1.91 SD = 0.78	M = 1.75 SD = 0.93	*p* = 0.13; Dz = −0.21
… are competent to detect signs of psychological stress in nurses.	M = 1.78 SD = 0.77	M = 2.32 SD = 0.85	*p* < 0.001; Dz = −0.20
… are competent to detect signs of psychological stress in paramedics.	M = 1.60 SD = 0.80	M = 1.52 SD = 0.97	*p* = 0.29; Dz = 0.15
… are competent to detect signs of psychological stress in students of their own professions.	M = 1.82 SD = 0.76	M = 2.00 SD = 0.81	*p* = 0.18; Dz = 0.15
… are competent to detect signs of psychological stress in supervisors of their own profession.	M = 1.68 SD = 0.80	M = 1.68 SD = 0.96	*p* = 0.004; Dz = −0.53
… address Students if they detect signs of psychological stress in them.	M = 1.75 SD = 0.95	M = 1.92 SD = 0.83	*p* = 0.004; Dz = −0.29
… address colleagues of their own hierarchical level if they detect signs of psychological stress in them.	M = 1.81 SD = 0.91	M = 2.05 SD = 0.83	*p* = 0.005; Dz = −0.29
… address supervisors if they detect signs of psychological load stress in them.	M = 1.03 SD = 0.76	M = 1.23 SD = 0.76	*p* = 0.02; Dz = −0.24
… take and accept feedback appropriately if they are addressed by another person in case of psychological burden or stress.	M = 1.71 SD = 0.73	M = 1.92 SD = 0.79	*p* < 0.001; Dz = −0.24

**Table 3 ijerph-19-16016-t003:** Results of paired samples t-tests with bootstrapping (bias-corrected and accelerated (BCa) based on 1000 samples) of differences between the assessment of physicians’ and paramedics’ competencies estimated by physicians. Dz shows effect Size according to Cohen’s Dz.

Item	Physician	Paramedics	*p*, Dz
… know the concept of “Second Victim.”	M = 1.63 SD = 0.97	M = 1.60 SD = 0.90	*p* = 0.70; Dz = 0.04
… are competent to detect signs of psychological stress in themselves.	M = 1.71 SD = 0.77	M = 1.92 SD = 0.81	*p* = 0.003; Dz = −0.32
… are competent to detect signs of psychological stress in physicians.	M = 1.91 SD = 0.78	M = 1.60 SD = 0.84	*p* < 0.001; Dz = 0.39
… are competent to detect signs of psychological stress in nurses.	M = 1.78 SD = 0.77	M = 1.57 SD = 0.81	*p* = 0.023; Dz = 0.25
… are competent to detect signs of psychological stress in paramedics.	M = 1.60 SD = 0.80	M = 2.08 SD = 0.87	*p* < 0.001; Dz = −0.57
… are competent to detect signs of psychological stress in students of their own professions.	M = 1.82 SD = 0.76	M = 1.83 SD = 0.86	*p* = 0.75; Dz = −0.03
… are competent to detect signs of psychological stress in supervisors of their own profession.	M = 1.68 SD = 0.80	M = 1.72 SD = 0.82	*p* = 0.38; Dz = −0.09
… address students if they detect signs of psychological stress in them.	M = 1.75 SD = 0.95	M = 1.90 SD = 0.87	*p* = 0.17; Dz = −0.14
… address colleagues of their own hierarchical level if they detect signs of psychological stress in them.	M = 1.81 SD = 0.91	M = 1.94 SD = 0.86	*p* < 0.001; Dz = −0.54
… address supervisors if they detect signs of psychological load stress in them.	M = 1.03 SD = 0.76	M = 1.48 SD = 0.86	*p* = 0.31; Dz = −0.54
… take and accept feedback appropriately if they are addressed by another person in case of psychological burden and stress.	M = 1.71 SD = 0.73	M = 1.77 SD = 0.75	*p* = 0.70; Dz = −0.11

## Data Availability

Data is available on request.

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
