# Peer review of "Self-Assessment and Learning Motivation in the Second Victim Phenomenon"

_ijerph, 2022, doi:10.3390/ijerph192316016_

Round 1
Reviewer 1 Report
Thank you for the invitation to review the manuscript ‘Self-Assessment and Learning Motivation in the Second Victim Phenomenon’.
The topic is of increasing importance in a healthcare system under unprecedented pressure, and this study presents a novel approach to the understanding of the second victim phenomenon.
I have two major concerns with the current manuscript. I) The response rate is less than 3% - this is very unusual and presents a great limitation. It should be stated clearly in the abstract and the conclusion should reflect this major limitation. II) The qualitative data analysis is insufficiently described – and this is where I focused my comments to contribute to the authors’ reflections:
The aim of the study is ‘to assess the presence of overconfidence concerning competencies in second victim recognition and management and their influence on participation in training programs’. I am not sure how the qualitative findings contribute to this assessment? Is this part of the research design appropriate to address the aims of the study? It is labeled Other analyses in the Result section – is it more of an appendix than actual main results? If not, perhaps it would be more appropriate to call it Qualitative findings and adjust/expand the aim of the study.
p. 16, ll. 1-4: This section lacks an in-depth description of the analysis process and which methodological orientation that underpins the study. It is unclear how the categories were derived from the data or to what extent contradictory data were considered (if relevant). Please explain the decontextualization process in a bit more detail.
p. 21, ll. 44-45: Please add the wording of the survey questions for the free-text boxes.
Is something missing after the ‘:’ in line 45? The sentence is difficult to understand.
p. 22+23 – Other analysis: Four themes have been constructed. The presentation of those appears to be at a preliminary stage with heavy listing of quotes and very little interpretation.
p. 22, ll 44-: The emotion model by Ekmans. The reference [24] does not mention a particular model? Please elaborate. And please insert a section in what you have labeled Qualitative variables to describe how the data analysis was conducted. A deductive approach is used for one theme – this must be unfolded, and the theoretical model described (by Ekmans?).
Discussion: The authors could further discuss the contribution the study makes to existing knowledge or understanding. How valuable is this research – please help the readers to translate this research.
p. 25, ll. 9-11: ‘Complete objectivity’ – does not make much sense to me in the context of self-reported data of any kind. Does not make the data less valid, but it would increase the academic quality of the manuscript to account for ontological and epistemological perspectives in this type of data generation.
It should be considered a limitation that only one person conducted the qualitative analysis.
Conclusion should reflect the major limitation of an extraordinary low response rate (less than 3%).
Author Response
Dear reviewer
Thank you for your comments that help us to improve our manuscript. We made following amendmends aside those suggested by the other 3 reviewers.
Thank you for the invitation to review the manuscript ‘Self-Assessment and Learning Motivation in the Second Victim Phenomenon’.
The topic is of increasing importance in a healthcare system under unprecedented pressure, and this study presents a novel approach to the understanding of the second victim phenomenon.
I have two major concerns with the current manuscript. I) The response rate is less than 3% - this is very unusual and presents a great limitation. It should be stated clearly in the abstract and the conclusion should reflect this major limitation.
This is an important issue. Thank you for this feedback. Online surveys are known to have response rates below 10% (e.g.: https://www.sciencedirect.com/science/article/pii/S2451958822000409). Additionally, the response rate might be even lower as the group addressed faces high workloads during the pandemic and thus response burden might be considered too high or newsletters and emails may be ignored. Nevertheless, our results should be considered “positivistic” but we agree that there might be more to be considered that should be assessed by quantitative and qualitative research. We included this limitation in the abstract and emphasized it in the discussion section as it already was a main point in the limitation section.
- II) The qualitative data analysis is insufficiently described – and this is where I focused my comments to contribute to the authors’ reflections:
The aim of the study is ‘to assess the presence of overconfidence concerning competencies in second victim recognition and management and their influence on participation in training programs’. I am not sure how the qualitative findings contribute to this assessment? Is this part of the research design appropriate to address the aims of the study? It is labeled Other analyses in the Result section – is it more of an appendix than actual main results? If not, perhaps it would be more appropriate to call it Qualitative findings and adjust/expand the aim of the study.
Thank you again for these comments. You are right. We re-labeled the section into “qualitative findings” and agree that it is not part of the main study. Thus, we reflected if these findings really needed to be reported on. We decided to include the qualitative findings (clearly to be acknowledged as secondary data) as they depict the situation and give options for further hypothesis generation and ongoing research.
- 16, ll. 1-4: This section lacks an in-depth description of the analysis process and which methodological orientation that underpins the study. It is unclear how the categories were derived from the data or to what extent contradictory data were considered (if relevant). Please explain the decontextualization process in a bit more detail.
We described the qualitative analysis in more detail and added some elements about the process itself. This data is purely secondary and did not reach data saturation (we added this in the discussion section). Therefore, we agree with the reviewer and declared the section as “qualitative findings” that may contribute for robust qualitative research in this field.
- 21, ll. 44-45: Please add the wording of the survey questions for the free-text boxes.
We revised this section and described the two free-text entries in more detail.
Is something missing after the ‘:’ in line 45? The sentence is difficult to understand.
No this is a typo (“.” Instead of “:”). It was revised.
- 22+23 – Other analysis: Four themes have been constructed. The presentation of those appears to be at a preliminary stage with heavy listing of quotes and very little interpretation.
As we did not reach data saturation this is true. Therefore, it is a good recommendation to add the term “preliminary”. Further, we emphasized the need for deeper qualitative research in this topic with a more robust qualitative approach that does not only rely on free text entries but perhaps video-taped interviews. We pointed this out in the discussion/limitation section.
- 22, ll 44-: The emotion model by Ekmans. The reference [24] does not mention a particular model? Please elaborate. And please insert a section in what you have labeled Qualitative variables to describe how the data analysis was conducted. A deductive approach is used for one theme – this must be unfolded, and the theoretical model described (by Ekmans?).
The reference is a consensus described/published by Ekman. We revised the analysis section and some points in the results section
Discussion: The authors could further discuss the contribution the study makes to existing knowledge or understanding. How valuable is this research – please help the readers to translate this research.
Thank you! We added the context of this study in the conclusion section as it might help SVP researchers, didactic researchers, supervisors and assisting persons
- 25, ll. 9-11: ‘Complete objectivity’ – does not make much sense to me in the context of self-reported data of any kind. Does not make the data less valid, but it would increase the academic quality of the manuscript to account for ontological and epistemological perspectives in this type of data generation.
We deleted this sentence.
It should be considered a limitation that only one person conducted the qualitative analysis.
We considered this.
Conclusion should reflect the major limitation of an extraordinary low response rate (less than 3%).
We added this point to the conclusion.
Reviewer 2 Report
In general the paper is interesting. The topic is the relationship between Self-Assessment and Learning Motivation in the Second Victim Phenomenon.
REVIEWS
Introduction
I recommend citing other European studies as well, a few examples:
https://pubmed.ncbi.nlm.nih.gov/28534429/
https://pubmed.ncbi.nlm.nih.gov/30921046/
https://pubmed.ncbi.nlm.nih.gov/27373579/
https://pubmed.ncbi.nlm.nih.gov/27378493/
https://pubmed.ncbi.nlm.nih.gov/27106771/
Page 2
Lines 16-21
I suggest to write:
In this theory motivation can be divided into intrinsic motivation “I want to learn” and extrinsic motivation to be partitioned into “identified regulation” “I have to learn, because it is my duty to do so”, extrinsic regulation “I must learn because another person tells me to do so” and amotivation “I am not interested in learning”. A tool to assess these four motivation entities in learning is the SIMS (Situational Motivation Scale) originally developed for learning in physical exercise [15].
Figure 1 is well done. I recommend removing the "" in the cells. We know they are text citations, but in the figures they are not needed
Page 3
Lines 20-27
This part is unclear, so don't put it in the introduction section, but I suggest to write it in the methods section.
Methods
Lines 30-31
We designed and conducted a cross sectional study in German language on overconfidence and learning motivation.
Please write the approval number of the ethics committee at the end of the manuscript.
Page 4
You did not report who constructed the questionnaire and whether it was validated (face validity, pilot test, internal consistency of questions etc.)
The table also contains results, so it should not be put in this session (results should be put in the results). The questionnaire should be divided by the results.
Also, the contents of the questionnaire in full can be attached, not here in such a long table
Page 16
More description of the qualitative analysis is needed.
Results
I found the results very complex to read. But it is also true that there are many variables in the study.
In the table 2 write nurses, in the table 3 write paramedics. It’s best to use the same word.
The tables 2 and 3 are very broad (also graphically should be harmonized). I suggest describing more in the text and making tables more summarizing and also smaller in size.
References
It is best to double-check all citations in order to adhere to the Vancouver style
Note: you will see that I have written very few comments to the results and discussion sections, this is because I cannot fully understand them, it is too much data and confusing. In all sincerity, I had difficulty reading the analysis. I think the study needs to be made more comprehensible so that the considerable work done is not lost
Author Response
Dear reviewer
Thank you for your comments that help us to improve our manuscript. We made following amendmends aside those suggested by the other 3 reviewers.
I recommend citing other European studies as well, a few examples:
https://pubmed.ncbi.nlm.nih.gov/28534429/
https://pubmed.ncbi.nlm.nih.gov/30921046/
https://pubmed.ncbi.nlm.nih.gov/27373579/
https://pubmed.ncbi.nlm.nih.gov/27378493/
https://pubmed.ncbi.nlm.nih.gov/27106771/
We added some of these very useful references to the introduction section
Page 2 Lines 16-21
I suggest to write:
In this theory motivation can be divided into intrinsic motivation “I want to learn” and extrinsic motivation to be partitioned into “identified regulation” “I have to learn, because it is my duty to do so”, extrinsic regulation “I must learn because another person tells me to do so” and amotivation “I am not interested in learning”. A tool to assess these four motivation entities in learning is the SIMS (Situational Motivation Scale) originally developed for learning in physical exercise [15].
Thank you. We removed the brackets and the typo.
Figure 1 is well done. I recommend removing the "" in the cells. We know they are text citations, but in the figures they are not needed
We removed the “”
Page 3
Lines 20-27
This part is unclear, so don't put it in the introduction section, but I suggest to write it in the methods section.
This part is part of the rationale and hypothesis of the study. As none of the other 3 reviewers recommended a transfer to methods, we did not change this at this stage of the review. Please provide further feedback if the demand is still existent.
Methods
Lines 30-31
We designed and conducted a cross sectional study in German language on overconfidence and learning motivation.
Please write the approval number of the ethics committee at the end of the manuscript.
The ethical committee decided on 21st of January 2022 that no votum is needed. Therefore, there is no approval number after the consultation.
Page 4
You did not report who constructed the questionnaire and whether it was validated (face validity, pilot test, internal consistency of questions etc.)
We added information about the transfer of the validated instrument from two other studies on overconfidence and learning motivation. In-Study validation was assessed, too (see page 32)
The table also contains results, so it should not be put in this session (results should be put in the results). The questionnaire should be divided by the results.
Also, the contents of the questionnaire in full can be attached, not here in such a long table
We moved the tables to the bottom of the document.
Page 16
More description of the qualitative analysis is needed.
We added more information about the qualitative findings and the coding process.
Results
I found the results very complex to read. But it is also true that there are many variables in the study.
In the table 2 write nurses, in the table 3 write paramedics. It’s best to use the same word.
The tables 2 and 3 are very broad (also graphically should be harmonized). I suggest describing more in the text and making tables more summarizing and also smaller in size.
We corrected that typo. The tables were formatted by MDPI . Further we added more analytic information in the results section for clarification.
References
It is best to double-check all citations in order to adhere to the Vancouver style
Note: you will see that I have written very few comments to the results and discussion sections, this is because I cannot fully understand them, it is too much data and confusing. In all sincerity, I had difficulty reading the analysis. I think the study needs to be made more comprehensible so that the considerable work done is not lost
We re-arranged the tables and figures (except figure 1) to the bottom of the document. Thus, division of the result section does not longer occur and might help to better understand the data provided. The analysis focusses on the hypothesis H1 to H3 with a minimum of statistics needed. As the manuscript might be of value to medical educators as well, we provided both p-values and Cohens D values.
Reviewer 3 Report
Dear Authors,
please refer to the attached file.
Kind regards,

Author Response
Dear reviewer, thank you for your valuable support for other study. we made the following amendmends.
Dear Authors,
After carefully reading your manuscript, I think the study was well conducted. My
congratulations for your efforts in scientific inquiry.
Thank you very much!
However, I think that for a full evaluation of your results and to improve the reader
understanding of your work the paper needs some elucidations, which I believe you can
properly address in an easy manner.
1) In materials and methods, I think that some further elucidation about the division of
the participants into “experts”, “recruitables” and “unaware” groups is necessary,
being it a key criterion.
We added a paragraph to the methods section addressing this issue.
Also, Table1 is missing the legend either in the first row or first column: I suppose M stands for Mean and SD for standard deviation, but I cannot be sure of it.
We corrected this.
Also, I think it is necessary to explicitly describe the scale used for every
question: again, I suppose that for the majority of items it is a Likert scale, but I was
unable to find this information.
This is correct. We added a section in the Methods section referring to the methodology in previous publications using the same instruments. Additionally, we added it in the description of the survey (Table 1).
2) Regarding Table 1, I suggest putting the German translation of the questionnaire into
supplementary materials, to improve readability.
We removed the German translation.
3) I strongly suggest to improve Figure 2 and 3 quality, and to reconcile the font
typology and size.
We reformatted this
4) I would explicitly specify that Dz is the Cohen effect size in the footnote or caption of
the tables.
We added it to the tables.
Reviewer 4 Report
Congratulations on the idea of such an original and very rarely studied area. Very good research tools - qualitative and quantitative. It is a pity that the remaining staff, e.g. nurses, were not examined at the same time. It also points to the need to search for/identify supervisors/tutors in a professional group and their continuous training, not only on paper – which is a very good approach to changing the approach to SVP.
Author Response
Dear reviewer
Many thanks for your comments on our manuscript that helps us to improve it for further consideration. We made the following amendments
Congratulations on the idea of such an original and very rarely studied area. Very good research tools - qualitative and quantitative. It is a pity that the remaining staff, e.g. nurses, were not examined at the same time.
Thank you very much. We are convinced that this grants opportunities for further research. We added it to the discussion section as this point addresses some issues of another researcher.
It also points to the need to search for/identify supervisors/tutors in a professional group and their continuous training, not only on paper – which is a very good approach to changing the approach to SVP.
Round 2
Reviewer 1 Report
Thank you for your replies to my comments and concerns. Most of them are addressed adequately.
I still have a few questions though.
Previous review:
- 21, ll. 44-45: Please add the wording of the survey questions for the free-text boxes.
We revised this section and described the two free-text entries in more detail.
è I am unable to find the wording, please add the page and line number. On page 22 you have added: 'Altogether 36 free-text entries could be analysed on the two questions in what situations participants experienced SVP and what happened after that and whether participants may have further suggestions for the survey or want to report on further experiences with SVP.' What is the exact wording? How many entries for each question?
- 22, ll 44-: The emotion model by Ekmans. The reference [24] does not mention a particular model? Please elaborate. And please insert a section in what you have labeled Qualitative variables to describe how the data analysis was conducted. A deductive approach is used for one theme – this must be unfolded, and the theoretical model described (by Ekmans?).
The reference is a consensus described/published by Ekman. We revised the analysis section and some points in the results section.
è It is still unclear what the model (or consensus??) by Ekman consists of and what it adds to the analysis.
Discussion: The authors could further discuss the contribution the study makes to existing knowledge or understanding. How valuable is this research – please help the readers to translate this research.
Thank you! We added the context of this study in the conclusion section as it might help SVP researchers, didactic researchers, supervisors and assisting persons.
è You have added this to the conclusion, not the discussion. Instead of providing suggestions for who may benefit from these results, I miss a clear translation of what the results are and how they may be used.
And finally; I agree with the other reviewers that this paper is rather difficult to read. There is a lot of (implicit) information. My suggestions are: 1) Provide a box of “what this paper adds” with short bullets explaining your findings, and 2) Use a professional language editing service to eliminate non-English terms and terminology that may add to the confusion of the reader.
Author Response
Dear Reviewer
thank you again for your time and comments on our study
we hope to have adressed all missing issues:
Reviewer #1 (Round 2)
- 21, ll. 44-45: Please add the wording of the survey questions for the free-text boxes.
I am unable to find the wording, please add the page and line number. On page 22 you have added: 'Altogether 36 free-text entries could be analysed on the two questions in what situations participants experienced SVP and what happened after that and whether participants may have further suggestions for the survey or want to report on further experiences with SVP.' What is the exact wording? How many entries for each question?We changed the paragraph as follows ad added the exact wording after translation:
Altogether 36 free-text entries on Q 57 and 26-free-text entries on Q58 could be analysed in what situations participants experienced SVP and what happened after that (Q57: In what situations did you experience Second Victims phenomena? What were the results of that?") and whether participants may have further suggestions for the survey or want to report on further experiences with SVP (Q58: “Do you have any suggestions, special experiences or comments about the Second Victim Phenomenon?”. Qualitative entries included 1933 words for both questions.
22, ll 44-: The emotion model by Ekmans. The reference [24] does not mention a particular model? Please elaborate. And please insert a section in what you have labeled Qualitative variables to describe how the data analysis was conducted. A deductive approach is used for one theme – this must be unfolded, and the theoretical model described (by Ekmans?).
The reference is a consensus described/published by Ekman. We revised the analysis section and some points in the results section.
It is still unclear what the model (or consensus??) by Ekman consists of and what it adds to the analysis.
We briefly explained the model /consensus in the results section and the reason why we used it and added a section in the discussion section to point out potential for future studies.
“In this second theme and after recognizing high emotional load we used a deductive approach according to the consensus on emotion described by Paul Ekmans (29): In this scientific consensus about human emotions, these are divided into the five basic emotions anger, fear, disgust, enjoyment and sadness with different intensity. We chose this deductive approach for better comparability with other research in medicine and the established use of coping mechanisms and communication techniques (e.g. mnemonics) addressing emotions known to physicians and that may not only help to support patients but colleagues too (32, 33).”
“A further finding of our analysis is that emotions like anger and anxiety may play a significant role in SVP indicating the need for implementation of briefing, debriefing (40) counselling or peer-support (41) with potential use of established communication strategies like the NURSE or SPIKES mnemonics for patients(32, 33) that may have to be evaluated in future studies for the use on colleagues.”
Discussion: The authors could further discuss the contribution the study makes to existing knowledge or understanding. How valuable is this research – please help the readers to translate this research.
Thank you! We added the context of this study in the conclusion section as it might help SVP researchers, didactic researchers, supervisors and assisting persons.
You have added this to the conclusion, not the discussion. Instead of providing suggestions for who may benefit from these results, I miss a clear translation of what the results are and how they may be used.
We added a new paragraph in the discussion on this issue. We hope it is addressed adequately:
Translation of the results
The results show either overestimation and overplacement effects in different groups of physicians, but not clinical tribalism effects. Further we detected emotional responses that are in line with preceeding studies on psychosocial effects auf traumatic events in health care emphasizing the need for educational programs on SVP and the establishment of self-care and self help in case of SVP and for communication strategies for peers and supervisors in case of strong emotional reaction before professional help can start: probably a “First Aid concept on Second Victims and Wounded Healers” directly on scene. Further we were able to show that the distinction in the three learning groups as shown in hand hygiene was not that prominent in SVP, possible due to the “novelty” of the term and concepts and low frequency in trainings on this issue. Thus, learning motivation may be high in “naïve” providers when hospitals and institutions are starting programs, but hypothetically may be followed by a “split-up” and more polarization into the three groups with creation of an amotivational group not easily motivated to attend training to acquire or maintain proficiencies in SVP recognition and support. Future studies should concentrate on this issue after a certain time the implementation of SVP programs or even better by longitudinal observation of self-assessment and learning motivation.
And finally; I agree with the other reviewers that this paper is rather difficult to read. There is a lot of (implicit) information. My suggestions are: 1) Provide a box of “what this paper adds” with short bullets explaining your findings, and 2) Use a professional language editing service to eliminate non-English terms and terminology that may add to the confusion of the reader.
Ad 1) sticking to the MDPI guidelines we added a short paragraph below the rationale to summarize the benefits to readers of the manuscript:
This manuscript provides readers with information on overconfidence effects about SVP Management, learning motivation and their intercorrelation. Additionally, it evaluated the existence of different learning types in SVP that might have to be addressed differentially by educators. Further it provides readers with additional quantitative and qualitative data on SVP contributing to existing evidence and context to prior studies.
Ad 2) Our native speaker again re-read and corrected the paper for better understanding. Prior we eliminated all German terms. However, we agree that there is terminology from different interacting scientific sectors and research, such as Second Victim Phenomena, Medical Didactics and Heuristics. We checked this again and are convinced that our research at the interface of these disciplines demands terminology from all sectors and preceding studies that concentrated on this issue.
Reviewer 2 Report
Dear authors,
The study is very complex and for some it will be difficult to read. But I believe that the effort made should be rewarded and that the paper can now be published
Author Response
We thank reviewer 2 for his comment on our study and the endorsement for publication.